# Orbital and millennial-scale forcing of the Patagonian Ice Sheet throughout the Last Glacial Cycle

Andrés Castillo-Llarena [1] ✉, Matthias Prange [1] & Irina Rogozhina[2,3,4]

During the Last Glacial Maximum (23,000 to 19,000 years ago), the Patagonian Ice Sheet covered the central chain of the Andes between 38 °S and 55 °S. The paleoclimatic evidence from Patagonia and New Zealand suggests that the maximum glacier expansion of the Southern Hemisphere mid-latitudes was desynchronized with the Northern Hemisphere glacial history. Here we present numerical simulations of the Patagonian ice sheet throughout the Last Glacial Cycle. Our analysis suggests that the Patagonian ice sheet had two main periods of advance, during the Marine Isotope Stage 4 and late Marine Isotope Stage 3, experiencing inter-millennial scale variability. We show that the Patagonian Ice Sheet long-term evolution can be attributed to changes in the integrated summer insolation, which combines the summer duration and insolation intensity and has an obliquity-like periodicity. We further suggest that this metric also modulated the behaviour of glaciers over the entire Southern Hemisphere mid-latitudes.

During the last glacial period, particularly during the Last Glacial Maximum (LGM, ~23,000 to 19,000 years ago[1]), the Earth hosted extensive ice sheets beyond those in Antarctica and Greenland. Notably, North America, northern Europe, and Patagonia were covered by the North American Ice Sheet complex, the Eurasian Ice Sheet complex, and the Patagonian Ice Sheet (PIS), respectively[2]. As the waxing and waning of ice masses is largely controlled by variations in temperature and precipitation, past ice sheets provide important information on past climate changes.

The PIS extended between approximately 38°S and 55°S, and recent geochronological advances have refined our understanding of its extent and chronology[3]. As such, studies of the evolution of the PIS provide unique insights into the Southern Hemisphere mid-latitudes climate system, for which there is a notorious lack of proxy information[4]. At the LGM, the PIS covered an estimated area and volume comparable with the size of the former British-Irish Ice Sheet that covered the British Isles during the LGM and was part of the Eurasian Ice Sheet complex[3,5,6]. Notably, based on geochronological reconstructions, it has been proposed that the PIS reached its

maximum extent at around 35 ka, during the Marine Isotope Stage (MIS) 3. Relatively stable conditions followed until 30 ka when it started to retreat until a temporary stabilization at around 20 ka before the final deglaciation[3]. Although the PIS extension over the last deglaciation has been relatively well constrained using moraine dating and glacial erratics, large uncertainty remains in both the pre-LGM evolution and the dynamical processes of the ice sheet related to millennial-scale variability, which are not resolved by geochronological reconstructions[3,7–9].

The temporal evolution of the PIS, in particular its asynchronous history when compared to the ice masses in the Northern Hemisphere during the global LGM, is still a matter of debate[4,10–12]. In general, paleoclimate proxy reconstructions suggest an extended LGM in the Southern Hemisphere, where the onset of maximum glacial conditions commenced from about 35 ka, even though this pattern is subject to spatial heterogeneity and millennial-scale variability[13]. The asynchronous timing of the Southern Hemisphere mid-latitude ice fields and glaciers, and particularly of the PIS, has been linked to the latitudinal migration and strength of the southern westerly winds[10,14,15] as well as

[1]MARUM - Center for Marine Environmental Sciences and Faculty of Geosciences, University of Bremen, Bremen, Germany. [2]Centro de Estudios Avanzados en Zonas Áridas (CEAZA), La Serena, Chile. [3]Department of Geography, Norwegian University of Science and Technology, Trondheim, Norway. [4]University of La Serena, Faculty of Engineering, La Serena, Chile. ✉e-mail: acastillollarena@marum.de

to the expansion of the Antarctic sea-ice, suggesting a dominant role of obliquity[16]. More recently, the equatorial migration and strengthening of the southern westerly winds, linked to global cooling and a steeper meridional temperature gradient, have been proposed as main drivers for the glacial expansion, as well as interhemispheric teleconnections[4].

Simulating the spatio-temporal evolution of the PIS over the Last Glacial Cycle (LGC, ~120,000 years ago to present day) is notoriously difficult due to the lack of continuous climate forcing fields[17]. However, a recently published alkenone-based sea surface temperature (SST) record from marine sediment core MR16-09 PC03 provides us with an opportunity to reconstruct the Patagonian climate over the past 140 ka as the core is situated about 150 km offshore the Taitao Peninsula in southern Chile/central Patagonia (Fig. 1), and has a temporal resolution high enough to resolve millennial-scale variability[9].

Here we present results from transient model simulations of the PIS throughout the complete LGC using the numerical ice sheet model SICOPOLIS[18] and making use of the MR16-09 PC03-based climate reconstructions. To this end, we employ a glacial index approach using two different alkenone-based temperature proxies $U^K_{37}$ and $U^{K'}_{37}$. The two indices differ in alkenone C37:4, which occurs mainly in cold regions. Both alkenone records show strong orbital-scale changes, suggesting glacial-interglacial sea surface temperature differences between 5 and 8 °C (Fig. 1) and reproducing coherent chronologies with the glacial histories of New Zealand and South America, suggesting the presence of large-scale common mechanisms[9,10,14,19]. At the core site, the choice of the index is critical, in particular since millennial-scale variability associated with cold Antarctic stadials is well pronounced in the $U^K_{37}$ record, but almost lacking in the $U^{K'}_{37}$ record (Fig. 2)[9]. Therefore, the use of both proxy records in our study allows us to elucidate the role of millennial-scale Antarctic cold stadials in the

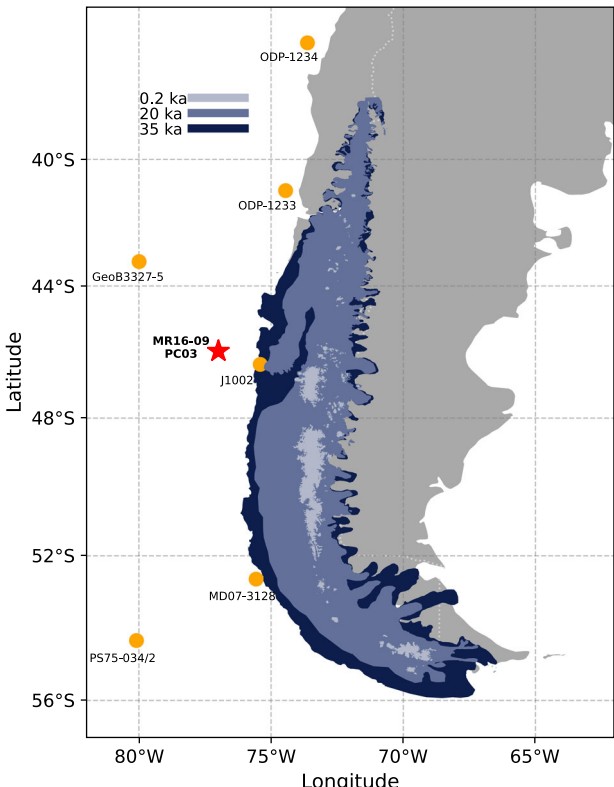

**Fig. 1 | Study zone.** PATICE geochronological reconstruction of the Patagonian Ice Sheet at 35 ka, 20 ka, and 0.2 ka are shown[3]. The position of the offshore record MR16-09 PC03 is shown with a red star[9]. The position of the Offshore records used to compare the results is shown in orange circles. Present-day continental margins are shown for reference.

dynamics of the PIS, even though Hagemann et al.[9] argue that $U^K_{37}$ is preferable at the core location.

The goal of this study is to explore the spatio-temporal history of the PIS throughout the LGC, assessing the timing and dynamics of glacial advances and retreats. In particular, we investigate the role of millennial-scale climate variability in forcing and response and revisit the orbital forcing mechanisms of the PIS.

## Results

The simulation forced by the $U^K_{37}$ record suggests an ice inception at around 115 ka (Fig. 2b). During the MIS 5, the ice conditions remained relatively stable, entering a quasi-steady state. By approximately 70 ka, coinciding with the onset of MIS4, the model reconstructs a significant increase in the ice extent. The growth of the ice sheet during this period resembles the extent reached during the LGM. This growth is punctuated by a millennial-scale variabiliy, associated with two prominent cold pulses, which facilitated ice sheet expansion, with a notable increase around 65 ka (Fig. 2a). However, these pulses did not solely contribute to the ice sheet's overall growth but instead marked a plateau of sustained glaciation through the late MIS4 to early MIS3, stabilizing near 40 ka. During the late MIS3 and the MIS2, millennial-scale variability has then triggered cold pulses that have promoted the ice sheet growth, reaching the maximum ice extent during the LGC. During this period, the results suggest that the ice sheet has experienced waxing and waning rather than a smooth glacial history (Fig. 2b) as opposed to the earlier suggestions based on geological reconstructions[3].

The simulation driven by $U^{K'}_{37}$ reveals a distinctive trajectory, with rapid ice sheet growth during MIS5, reaching ice extent comparable to the LGM. Notably, around 95 ka, the model exhibits a marked and abrupt reduction in ice extent, followed by a rapid recovery at approximately 90 ka. This decrease is associated with a warming signal in sea surface temperature, which is much less pronounced in the $U^K_{37}$ record (Fig. 2a). During the MIS4, the simulation performed with $U^{K'}_{37}$ showcases quasi-stable ice extent behavior, with only minor fluctuations, and a nearly absent millennial-scale variability as opposed to $U^K_{37}$-based simulation. The impact of the millennial-scale variability on the ice sheet geometry and thickness is mainly over the north- and south-western margins as well as over the eastern margin in the vecinity of MR16-09 PC03 (Supplementary Fig. 6). A modest growth occurs during the MIS4-MIS3 transition (Fig. 2b), but during MIS3 and MIS2, the ice sheet remains in a relatively stable condition, again indicating that $U^{K'}_{37}$ does not capture millennial-scale variability during the MIS2, suggesting minimal variations in the ice sheet geometry.

In summary, the simulation with the $U^K_{37}$ forcing suggests two main periods of glacier advance, one during the MIS4 and another during the late MIS3-early MIS2, with millennial-scale variability throughout the interval with the maximum ice extent. The modeled PIS is mainly drained by fast-flowing outlet glaciers to the east, while even larger velocities can be found in the western ice margin (Supplementary Figs. 3 and 5). The velocities of ice streams that terminate in the Pacific Ocean range over 1000 ma⁻¹, whereas the velocities of land-terminating fast-flowing outlet glaciers at the eastern margins of the PIS are generally within 100 ma⁻¹. The ice divide mainly extends north-south along the Patagonian Andes between. On the other hand, the simulation performed with $U^{K'}_{37}$ shows less variability, representing the LGM as a phase of stability with limited dynamical shifts in the ice sheet configuration (Fig. 3).

## Discussion

Biomarker records from the core MR16-09 PC03 revealed phases of high terrigenous input occurring at millennial time scales (Fig. 2e), which have been interpreted to reflect increased ice discharge due to a growing PIS[9]. It was further suggested that these phases were associated with millennial-scale Antarctic stadials. The results of our

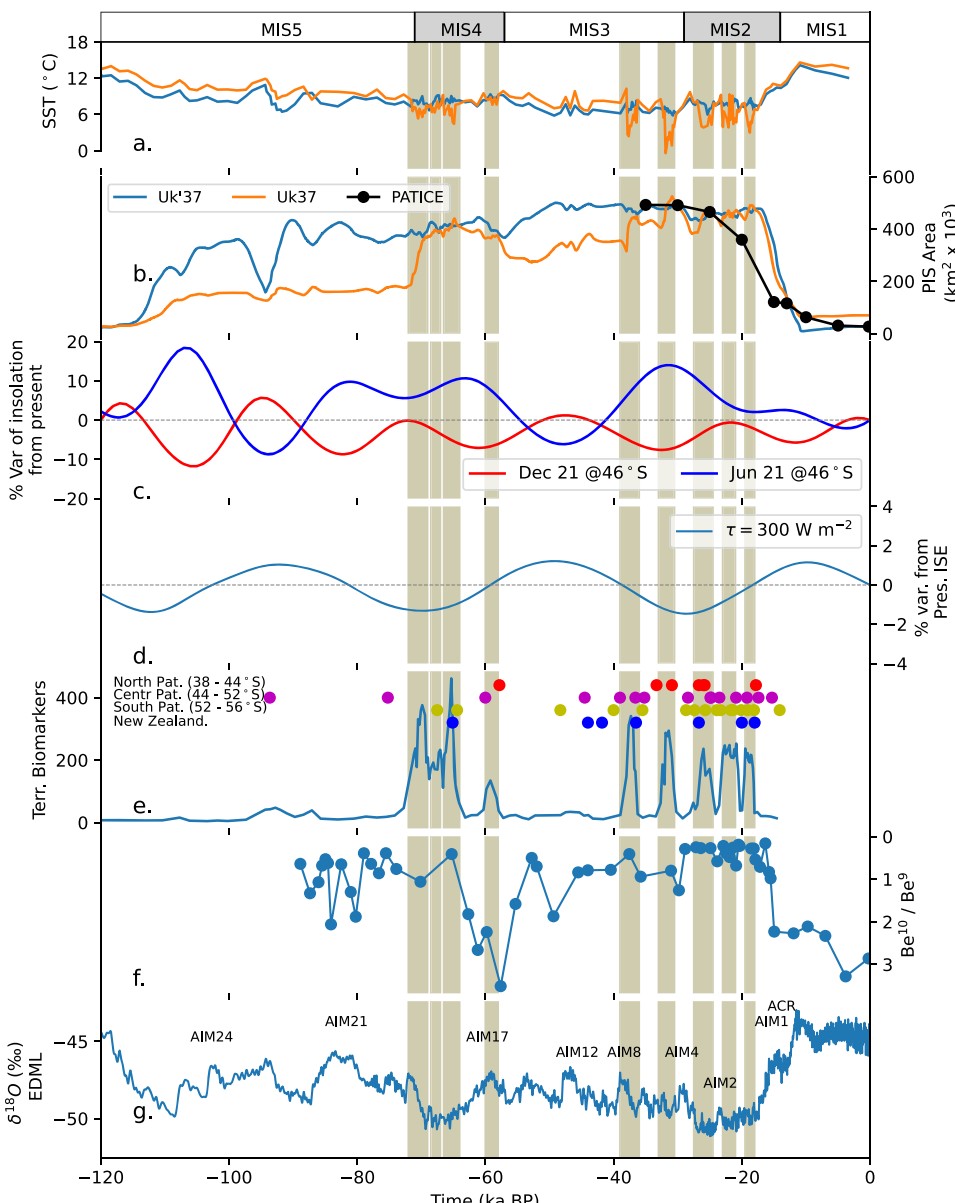

**Fig. 2 | Glacial history between 120 and 0 ka. a** Sea surface temperature (SST) reconstruction based on $U_{37}^{K}$ and $U_{37}^{K'}$ obtained from the offshore record MR16-09 PC03[9]. **b** Modeled Patagonian Ice Sheet (PIS) area by using the SST-based Glacial Index displayed in (**a**) and the PATICE reconstructed area of PATICE[3]. **c** Summer (21st of December) and winter (21st of June) insolation[38]. **d** Integrated Summer Energy (ISE) using a threshold of 300 Wm$^{-2}$ following Huybers[29] based on the insolation values of Laskar[38]. **e** Terrigenous biomarkers[9] and timing of glacial advance in New Zealand[20,39] and northern[40,41], central[42–44] and southern[7,45,46] Patagonia. **f** Beryllium (Be) isotope ratios for Site J1002[4]. **g** [18]O record from the EDML ice-core[30]. ACR Antarctic Cold Reversal, AIM Antarctic Isotope Maxima.

simulation forced by the $U_{37}^{K}$ record confirm the important role of cold stadials in driving the growth and variability of the PIS. Phases of PIS growth in this simulation coincide with phases of enhanced terrigenous input during the MIS4 and late MIS3 (Fig. 2b). This high-frequency variability is poorly captured by geological reconstructions of ice margins due to limited in situ constraints. Available reconstructions indicate relatively stable ice conditions despite the heterogeneous chronologies along the eastern margin of the PIS[3,10]. The glacial advances during the MIS4 and late MIS3 are closely aligned with the broader glacial history of New Zealand[20], East Falkland[21], South Georgia[22], and Marion Island[23], highlighting that large-scale climate mechanisms might have driven the PIS evolution during the LGC (Fig. 2e).

The $U_{37}^{K}$-forced simulation also shows strong coherence with the Be isotope record from Site J1002 (Fig. 1), close to the core MR16-09 PC03[4]. Both datasets illustrate similar glacial-interglacial patterns and

phases of the PIS growth (Fig. 2f). These similarities bolster the credibility of the $U_{37}^{K}$-based reconstruction in capturing the dynamics of the PIS. The onset of the glacial advance during the MIS4 is proposed to be earlier by Sproson et al.[4]. However, we associate those temporal discrepancies with the increase in the age model uncertainties, particularly during the proposed glacial advance around the MIS4.

Despite some overall agreement, the results of $U_{37}^{K}$ and $U_{37}^{K'}$ simulations diverge significantly during the MIS5 and MIS3 periods (Fig. 2b). While both records generally track the PIS glacial history, $U_{37}^{K}$ provide a better fit with both the MAR record of Hagemann et al.[9] and records of Sproson et al.[4], suggesting two main periods of glacial advance, rather than a quasi-stable glacial condition throughout the LGC. Specifically, during the MIS3-MIS2 transition, $U_{37}^{K}$ captures high variability phases that $U_{37}^{K'}$ does not, suggesting that $U_{37}^{K}$ offers more reliable insights into glacial dynamics, particularly when near offshore records are considered[8,9,24]. During the MIS5, our simulation with $U_{37}^{K'}$

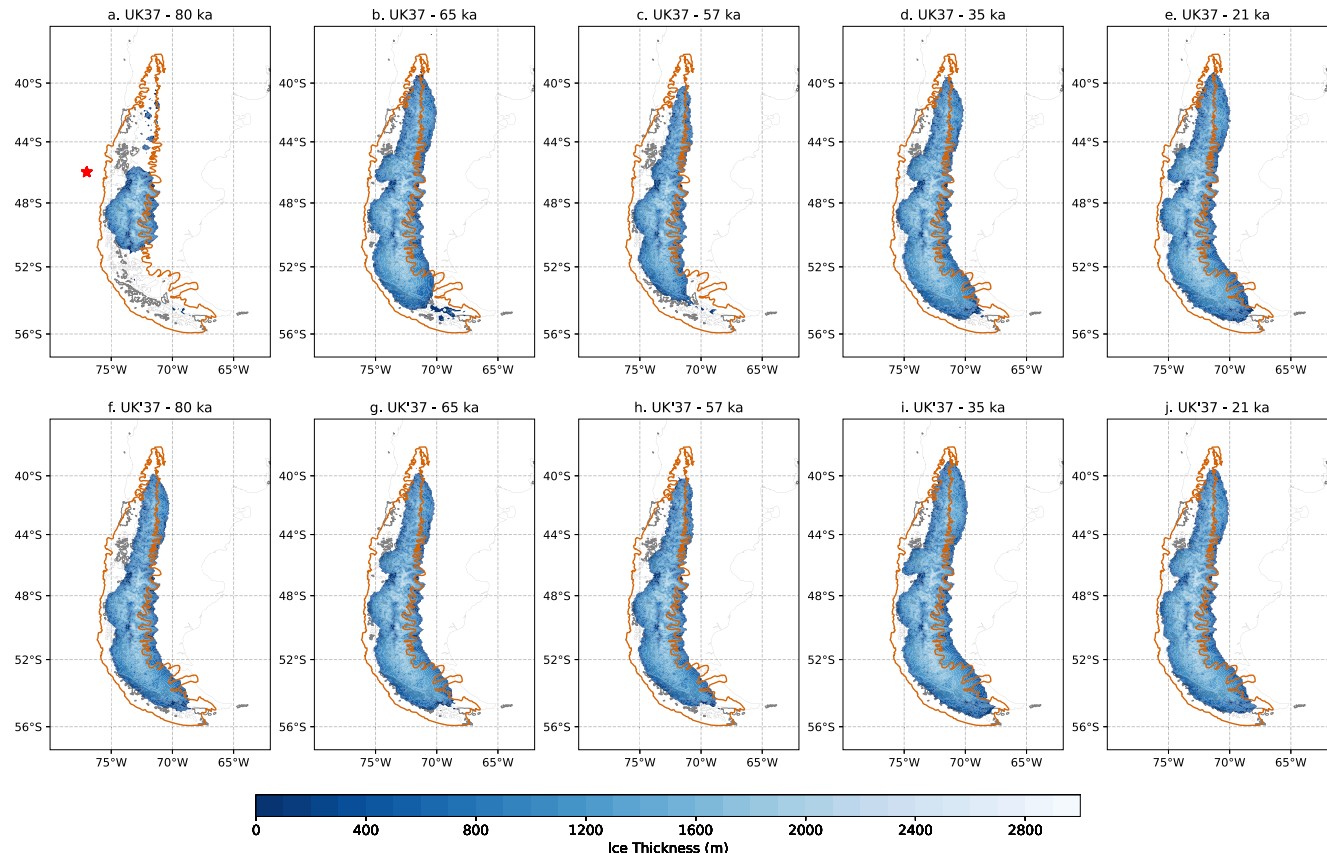

**Fig. 3 | Ice sheet geometry. a–e** Ice thickness modeled at 80, 65, 57, 35, and 21 ka by using the glacial index method based on $U^K_{37}$. **f–j** Ice thickness modeled at 80, 65, 57, 35, and 21 ka by using the glacial index method based on $U^{K'}_{37}$. Those snapshots are chosen as representatives of the Marine Isotope Stage (MIS) 5, MIS4, early MIS3, MIS3-MIS2 transition, and Last Glacial Maximum, respectively. Both proxies, $U^K_{37}$ and $U^{K'}_{37}$, were obtained from the offshore records MR16-09 PC03[9]. PATICE geochronological reconstruction of the Patagonian Ice Sheet at 35 ka is shown in orange[3]. Red star indicates the position of the offshore records MR16-09 PC03[9].

shows an earlier inception and growth than those with $U^K_{37}$, nearly to the LGM conditions (Fig. 2b). The glacial index method takes as reference values the pre-industrial conditions and the LGM. The millennial-scale variability captured by $U^K_{37}$ inserts an offset of 1.5 °C colder than $U^{K'}_{37}$ during the LGM, interpolating the conditions during the MIS5 as non-LGM-like conditions, unlike $U^{K'}_{37}$ that reconstructs conditions similar to the LGM.

Our results suggest that millennial-scale Antarctic cold stadials played a critical role in the PIS growth and variability (Fig. 2b, g). Increased ice discharge phases, as recorded offshore, are closely associated with Antarctic stadials[9]. These findings are consistent with previous studies linking southern mid-latitude glacial advances to fluctuations in southern westerly winds and oceanic fronts, often synchronous with the Antarctic cold phases[4].

Our simulation aligns well with the total area reconstructed by PATICE[3] for the MIS3-MIS2 transition (Fig. 2b). The PATICE dataset suggests that the maximum ice extent in Patagonia took place at around 35 ka, while a rapid deglaciation began at around 18 ka, which coincides with the simulation's portrayal of high variability and rapid ice loss during this interval (Supplementary Figs. 4 and 5). The ice sheet thickness modeled in this study is within the range of the existing literature[6,17,25,26]. However, while PATICE proposes a relatively smooth glacial history, our $U^K_{37}$-based results indicate inter-millennial fluctuations in the PIS extent during this time as was also suggested by Hagemann et al.[9]. Mismatches in the ice extent have been found not only in this research work but also in previous works, being largely attributed to the coarse resolution of the atmospheric forcing and the poorly represented topography[17]. Coarse and inaccurate orographic forcing in Patagonia induces wetter conditions and colder

temperatures over the leeside of the Andes[27], promoting ice expansion beyond its geologically reconstructed margin over its fasten domain[3].

We further suggest that local peak summer insolation cannot fully account for the PIS growth and demise during the LGC (Fig. 2a, b). The summer insolation in the southern mid-latitudes shows a relative maximum at the beginning of the MIS4 period, when the PIS experienced a glacial advance[4,9,10,20]. The winter insolation in the Southern Hemisphere mid-latitudes cannot explain the observed glacial advance history of the PIS either (Fig. 2c). Relative maxima can be found during MIS4 and late MIS3, while a minimum of insolation takes place during the early MIS3, preventing larger accumulation that would enable a PIS growth[28]. The insolation pattern alone contradicts the observed growth, indicating that other factors must have played a role in driving the glaciation.

Sproson et al.[4] proposed that summer duration serves as a more suitable orbital metric for understanding the PIS evolution. They argued that shorter summers in the Southern Hemisphere allowed for more extensive ice buildup, a factor not fully captured by peak summer insolation values. This approach better explains the timing and extent of PIS growth compared to insolation alone.

We propose that the integrated summer energy (Fig. 2d), which combines summer duration with the insolation intensity[29], provides an even more accurate metric for explaining PIS evolution (Supplemetary Fig. 1). The local minimum in integrated summer energy during the beginning of the MIS4 aligns well with the timing of the PIS growth, while the positive anomaly in early MIS3 helps explain the gradual disintegration of the ice sheet (Fig. 2b). Moreover, during the MIS3-MIS2 transition, a minimum in the integrated summer energy matches with the proposed maximum PIS extent over the LGC[3]. We suggest that

this metric offers a more nuanced understanding of the PIS behavior than either peak insolation or summer duration alone.

Existing modeling efforts of the PIS have concluded that the timings of maximum glacial advance and the consequent asynchronicity with the global mean behavior are not captured by Antarctic ice cores, and local proxy data is needed to capture the local variability[17]. However, the orbital-scale variability has imprints in different offshore records in the Pacific Ocean (Supplementary Fig. 2) and the EDML Antarctic ice core record[30] (Fig. 2g). A previous modeling study focusing on the PIS has shown that Antarctic ice core record reproduces a relative maximum of glacial conditions in broad agreement with the ISE, suggesting two main periods of glacial advance centered around the MIS4 and late MS3-MIS2[17].

Here, through a numerical modeling approach and its comparison with in situ data, we have investigated the PIS history throughout the LGC by performing transient simulations driven by the glacial index approach. The glacial indices were derived from offshore records (MR16-09 PC03) located 150 km offshore Chile at 46 °S in central Patagonia. Our findings suggest that the PIS had two main periods of advance: during the MIS4 and the MIS3-MIS2 transition. Southern Hemisphere peak summer insolation cannot explain the timing of the ice variability in Patagonia. Instead, periods of glacial advance are associated with low integrated summer energy. Combining the summer duration with the insolation intensity, the integrated summer energy has an obliquity-like variability throughout the LGC. In addition, the integrated summer energy pattern would also partially explain the earlier maximum ice extent not only of the PIS but also elsewhere in the Southern Hemisphere mid-latitudes. Furthermore, millennial-scale climate variability favored glacial advances and is related to inter-millennial variations in PIS extent.

## Methods
### Ice sheet model and climate forcing
We apply the open-source, three-dimensional, thermomechanical ice sheet model SICOPOLIS (SImulation COde for POLythermal Ice Sheets[18]) for the area between 80 °W and 62 °W and between 36 °S and 58 °S. The ice model setup was presented in a previous publication[17] and is briefly summarized here. Supplementary Table 1 contains an overview of the applied model parameters.

The inception and evolution of the PIS in our model are driven by the surface mass balance (SMB), which is calculated as the difference between the applied fields of accumulated precipitation and surface ablation. The latter is computed using a positive-degree-day (PDD) model[31], based on the given near-surface air temperature field. PDD parameters have been selected based on contemporary and paleo studies in the area[17,26,32]. Surface mass accumulation is assumed to depend on monthly precipitation and temperature fields. The transition between solid and liquid precipitation is linearly proportional to variations in air temperature with 0 °C to 2 °C limiting purely solid or purely liquid, respectively[33]. As the model domain surface evolves, discrepancies between the prescribed (fixed) topography used in the climate model snapshots and the dynamic one in SICOPOLIS are accounted for by implementing a near-surface air temperature lapse-rate correction of − 6.5 Kkm$^{-1}$. The precipitation changes by 7.3% for each degree Celsius of air temperature change[34]. Glacial isostatic adjustment of this bedrock produced by temporal variations in the ice mass load is accounted for through an elastic lithosphere-relaxing asthenosphere (ELRA) model[35] using standard parameter values. To isolate the atmospheric dynamics and the ocean dynamics, the model is set as such that it calves as soon as it reaches the coast.

Aiming to explore the glacial history of the PIS before the global LGM, we perform transient simulations of the PIS throughout the full LGC. To generate a climate state at any given model time, LGM and PI conditions from MPI-ESM1-2-LR have been used[36], representing peak glacial and interglacial conditions. Based on previous studies,

MPI-ESM1-2-LR stands as the best-fitted paleoclimate model in Patagonia for equilibrium and transient studies[17,26]. These two snapshots are then subject to a weighted interpolation following a glacial index approach with

$$GI(t) = \frac{X(t) - X_{PI}}{X_{LGM} - X_{PI}}, \quad (1)$$

where $t$ is time and $X$ is the SST derived from the offshore records[37]. The time-dependent climate-forcing fields are then given by

$$T(t) = T_{PI} + GI(t)(T_{LGM} - T_{PI}) \quad (2)$$

$$P(t) = P_{PI}\left(1 - GI(t) \cdot \left(1 - \frac{P_{LGM}}{P_{PI}}\right)\right), \quad (3)$$

where $T$ and $P$ represent the temperature and precipitation fields through time, respectively.

In the present study, the time-dependent weight GI is derived from the offshore sediment core MR16-09 PC03. The core covers the past 140 ka, being retrieved at around 46 °S, 77 °W from a water depth of 3082 m[9]. SST records were derived from two different alkenone-based proxies from the same core, $U^K_{37}$ and $U^{K'}_{37}$. The computation of GI uses the 19–23 ka mean of SST to define peak glacial conditions. Likewise, peak interglacial conditions are defined near the PI period by averaging over the last 3 ka. Each simulation is initiated from ice-free conditions.

### Calculation of integrated summer energy
The integrated summer energy[29] has been calculated using the daily mean insolation data from ref. 38. The integrated summer energy was calculated assuming $ISE = \sum_i W_i \cdot 86400$), with $W_i$ being the daily mean insolation and i equal to one when $W_i$ is greater than or equal to a threshold $\tau$ and equal to zero otherwise[29].

## Data availability
The data required to reproduce all figures in this study have been deposited in the Zenodo database under accession code zenodo.org/records/17131020. https://doi.org/10.5281/zenodo.17131020.

## Code availability
The SICOPOLIS base code of version 5.2 that was used for all simulations shown in this manuscript is available at Zenodo under zenodo.org/records/17131226. https://doi.org/10.5281/zenodo.17131226.

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

## Acknowledgements

A.C.L. acknowledges support from the Agencia Nacional de Investigación y Desarrollo (ANID) Programa Becas de Doctorado en el Extranjero, Becas Chile, for the doctoral scholarship. A.C.L. and M.P. acknowledge support from the PalMod project under grant numbers 01LP2315A and 01AP2304A. A.C.L. and M.P. acknowledge the support of Cluster of Excellence EXC 2077 ("The Ocean Floor—Earth's Uncharted Interface"). The authors thank Dr. Julia Hagemann for the forcing data and the

discussion about it. The authors thank the University of Bremen for funding this research article. This study focuses on Patagonia and includes contributions from a Chilean researcher and another researcher based in Chile, ensuring that regional perspectives are represented. Our author team brings together diverse cultural and national backgrounds, reflecting a commitment to inclusive and collaborative research.

## Author contributions

The original concept was conceived by A.C.L. and M.P. Experiments have been carried out by A.C.L. A.C.L., M.P., and I.R. contributed to the writing and reviewing processes.

## Funding

## Competing interests

The authors declare no competing interests.
