## [Transparent Peer Review file · Nature Communications]

Orbital and millennial-scale forcing of the Patagonian Ice Sheet throughout the Last Glacial Cycle

Corresponding Author: Mr Andrés Daniel Castillo-Llarena

Version 0:

Reviewer comments:

Reviewer #1

(Remarks to the Author)

Review of "Orbital and millennial-scale forcing of the Patagonian Ice Sheet throughout the Last Glacial Cycle"

Andrés et al present a modeling study of the Patagonian Ice Sheet (PIS) through the last glacial cycle that explores why the PIS reached its maximum likely before the global LGM, and to a lesser extent, why this is out of phase with Northern Hemisphere ice sheets. This study adds to an expansive body of literature over the last several decades that have all focused on 1) when were glaciers/ice sheets in the Southern Hemisphere expanded, 2) when did they retreat, 3) what forced these changes, and 4) how/why is this signature different from the Northern Hemisphere.

I am not an ice-sheet modeler, but I am a Quaternary paleoclimatologist/geochronologist who thinks about these questions above. As such, I found this paper quite interesting and it had me thinking about my own research and sifting through papers and furthermore, I think this manuscript is generally well written. I honestly cannot comment if this manuscript is "correct", but the methods seem sound, the discussion reasonable, and I do think it a worth addition to the broader discussion surrounding Southern Hemisphere glaciation. I only have a few suggestions and questions:

1) At first, I struggled a bit with "What is the main selling point here?", which frankly is probably not a good sign for a manuscript like this. I think in the end the main selling points are a) a scheme that uses integrated insolation as a driver of long-term change provides the best data-model fit, and 2) more traditional treatments of insolation are unable to explain the pattern of PIS glaciation. I encourage the authors to really emphasize integrated insolation. For example, its mentioned in the abstract, but the first thing I read (and thought was the main takeaway) was that the PIS advanced during MIS/3...to which my response was "we've known that for awhile."

2) Selection of sediment core MR16-09 PC03. This seems a bit arbitrary, and I do wonder about how reproducible these results are using other temp records. I strongly suspect this record was used because it's the longest record from the region and spans the last glacial cycle (and probably the most recent record?). But it may be worth exploring how other records affect the results. ODP sites 1233 and 1234 slightly north of the record used here also have alkenone SST records. I believe at least one of those also spans the entirety of the last glacial. How easily can these be incorporated. At the bare minimum, I do not think these records can be completely ignored....are they different/similar to MR16-09 PC03? Should other records be plotted on a figure as well?

3) Like #2, I find it hard to believe I just read an entire Southern Hemisphere glaciation paper without seeing a single ice-core record plotted, let alone used. The authors routinely mention that "millennial-scale Antarctic cold stadials played a critical role...." I think the authors need to at the bare minimum show these on a figure somewhere. If I'm playing devil's advocate, what would happen if you simulated the last glacial cycle using a temp record from Antarctica, or at least WAIS (realizing that WAIS does not go back 140ka)? Has this been done before? Maybe what I'm also getting at here is how do the authors simulations compare to previous efforts. Since this is Nature Communications I think you have a little more leeway with article length.

4) I like how the authors made a decent attempt to bring in the geologic record. I think many modeling groups often don't even bother, so this is a nice touch. My only comment on this aspect would be how the authors evaluate model-data success or mismatches. There are a lot of references to how the author's efforts *broadly* do a good job at reproducing the geological

record of PIS change, but it is a bit tough to gauge from the text and figures. I will say that having the 35-ka outline in Figure 3 is very helpful although I was wondering a bit about the slight mismatch between model output and the geologic record at 35 ka. Might be nice for the authors to comment on that. Also, why does the model output appear shifted east at 35 ka? Or maybe another way to put it is that it undershoots ice on the west side but overshoots on the east side. I think this paper would be stronger if the comparison between model output and the geological record were a bit more robust, maybe there is some way to quantify this. The most obvious targets/benchmarks are MIS 4 (PIS at max near max as constrained by geologic record) and MIS 2 where there are numerous dated moraines.

Reviewer #2

(Remarks to the Author)

This manuscript is one of the first to attempt to model the Patagonian Ice Field / Ice Sheet through the Last Glacial Period using a new ocean core that provides an alkenone-based sea surface temperature record (MR16-09 PC03) from Hagemann et al. (2024). The ability to use such a record is really interesting and fascinating, as such attempts have previously not been able to do so. I do commend the manuscript for attempting to do so and being at the forefront of modelling the Patagonian Ice Field using such recent and new data.

Noteworthy results

- Modelling the Patagonian Ice Field through the Last Glacial Period itself is noteworthy
- Multiple advances of the Ice Field both within MIS 4 and 3 that are varied due to inter-millennial variability
- That, using the records provided, the simulations agree with PATICE (Davies et al., 2020), but disagree with a smooth deglaciation but rather varied deglaciated with interannual fluctuations
- Summer and winter isolation do not provide a clear explanation for ice variability, but more closely follow summer energy.

Does the work support the conclusions?

The evidence provided in Figure 2, the model output, and the comparison with other records provides clear evidence for the conclusions presented within the paper. Would have been nice to have seen more of a comparison with more outlines of the Patagonian Ice Sheet, to see it through different timesteps, and if the model with the new SST record could match more than just the 35ka transition.

Flaws in data analysis:

While the output itself of the ice extent and thickness are valuable in and of themselves, I think more could be said from your model which you have negated to do so. It would be really interesting and useful to understand what is happening with the velocity fields, how the ice is evacuating itself through the ice sheet. Does this change as the ice sheet involves through the Last Glacial Period? Is the thickness realistic, and does it match what other models have done over Patagonia for the LGM for example? While it may capture 35ka well, does it actually match other time periods well?

Methodology:

This modelling is a continuation of a modelling study from the same authors over Patagonia (Castillo-Llarena et al., 2024). While much of the methodology is similar, there are certainly some differences which should be noted in the methodology. For example, how the input data was made and set up for the model. What are the parameters used in the model? Any differences from the default parameter values? This could be in the form of a table so we know the parameters varied, as from reading the former paper before this, there is also no clear indication of parameter values. Also, if there was any additional parameters or modules used to enable the model to be ran through to completion. At the moment, I do not think there is enough for replication purposes.

Attached is also a commented version of the manuscript for minor points.

Overall comment and review verdict:

I personally think the amount of information that could be extracted from such a study is not inline with the reduced amount of information a nature paper allows the author. While the study and findings are some of the first to have been conducted over Patagonia, the outputs need to be described in-depth, with further information also to be presented in the methodology. The study itself is highly commendable and a pleasure to read, but I do not see Nature Comms being the best fit for such a paper. If the authors wish to resubmit to another journal, I would be more than happy to be put down as a reviewer to provide further feedback.

Reviewer #3

(Remarks to the Author)

This study produces a new reconstruction of the Patagonian Ice Sheet (PIS) across the last glacial cycle, extending a previous reconstruction which largely covered MIS3. The authors relate changes in their modeled PIS to integrated summer insolation and millennial-scale climate variability. The ice sheet reconstruction is forced by a continuous SST record over the past 140ky from the eastern Pacific offshore of Chile, which records variability in the western extent of the PIS. This provides insight into regional climate dynamics over a significantly longer period than previous nearby records. This SST record is used to construct a glacial index which guides the interpolation of glacial and pre-industrial climate states to be used as model forcing.

The results of this study provide important constraints on the history of the PIS beyond what was previously available and

reveals new millennial-scale dynamics within previously studied intervals. I congratulate the authors on the work that went into this study, and it will undoubtedly provide a great resource to gain further insight into the PIS.

My main points of discussion focus on the capability to meaningfully differentiate between the two forcings across MIS5, which in my view is weakened by the lack of independent verification and uncertainty in the forcing. I will discuss this below, along with the authors three primary findings which I take directly from the abstract:

(1) "Our analysis suggests PIS advances during the Marine Isotope Stage (MIS) 4 and late MIS3..." This finding is dependent on the absolute temperature of the Uk37 record, which is the primary difference, rather than high-frequency variability, between the two forcings across MIS5.

(2) "We show that millennial-scale climate variability played a role in the PIS dynamics..." This is an important observation for this study, as it shows that the model can sustain a dynamic PIS in-phase with millennial-scale climate variability.

(3) "...while the long-term PIS evolution can be attributed to changes in the integrated summer insolation" This is also an important observation. However, it is unclear if this observation is limited to this reconstruction or is a consistent feature across all of the Southern Hemisphere.

Overall, this study provides valuable new information surrounding the history of the PIS. A strength of this study is the identification of millennial-scale PIS variability across MIS3, which warrants further description. In my view, this paper should provide more conservative claims surrounding the MIS5-4 history and additional insight into the dynamics of newly observed millennial-scale variations.

Main Discussion Points:

1) My main discussion point is the impact of the of glacial index in the PIS reconstruction. This is particularly evident in the difference in extent during MIS5, where the reconstruction based on Uk'37 rapidly expands at ~115 kya. I presume this is due to the systematically lower temperatures in Uk'37 as compared to Uk37.

How well understood is the temperature uncertainty using these SST reconstruction methods? Neither this study, nor the data source Hagemann et al., 2024, discuss this. The difference between records is ~1.25-1.5oC, which is in the range of absolute uncertainty in comparable reconstruction methods. Based on the uncertainty in the SST's, can the different PIS reconstruction scenarios be differentiated?

While I agree with the author's interpretation that other lines of evidence (primarily the MAR/biomarker data of Hagemann et al., 2024 and the glacial advancement markers cited within Hagemann et al., 2024) suggest that the PIS reconstruction based on UK'37 can be excluded, I feel that discussion of uncertainty within the SST forcing and the sensitivity of the model to that forcing is a necessary discussion in order to justify an advancement across MIS4 rather than MIS5. As it stands, the results only suggest that PIS initiation occurred by ~115ka and reached its approximate LGM extent by at least the onset of MIS4.

2) For the observed millennial-scale variations in extent and volume, where do these changes manifest? I feel this is key source of information which is not discussed. Are these changes in extent isolated to specific regions, or are they well distributed across the PIS? This could be included as a supplemental spatial anomaly map.

How do the changes in this new reconstruction compare to other reconstructions like PATICE or that of Darvill et al., 2016? Does the difference in extent in Figure 3 in the 35ka panels imply this reconstruction underestimates the geological evidence?

3) In a recent study (Castillo-Llarena et al., 2024), Antarctic ice core data is used as a forcing over the last 70 kya. Has this been done for comparison over the last glacial cycle here? This could provide a necessary source of independent verification for the orbital-scale variability of the PIS.

The structure and magnitude across MIS3 is also consistent between the EDC forced reconstruction of Castillo-Llarena et al., 2024 and Uk37 shown here. Including a comparison between the two could provide insight into their similarities and differences across both millennial and orbital timescales. Furthermore, how do the other reconstructions of Castillo-Llarena et al., 2024 which utilize other SST records across MIS3/2 compare to the ones presented here? If the PIS advancement during late MIS3 is a regional phenomenon, is it expressed across the wider SST forcings and in their respective ice sheet reconstructions?

4) In a similar vein, is statement in lines 136-138: "The local minimum in integrated summer energy during the beginning of the MIS4 aligns well with the timing of the PIS growth, while the positive anomaly in early MIS3 helps explain the gradual disintegration of the ice sheet (Fig. 2b)." further supported by previous reconstructions of the PIS?

5) Does this ice sheet reconstruction support glacial sediment delivery to the site? The western margin is reduced as compared to previous reconstructions, with more significant advances across the eastern margin. Does this reduced western extent still result in marine terminating conditions, or ice extent across the continental margin as suggested by Hagemann et al? In other words, is the PIS reconstruction presented here consistent with the conditions necessary to reproduce the sediment forcing record?

Minor Comments:

Line 87-88: References Figure 2e here.

Line 99-101: Does the Be10/Be9 contradict the results of PIS advancement at the start of MIS4? The Be10/Be9 is indicative of terrigenous sedimentation beginning at ~80kya, with oscillations preceding it. The record across MIS5 is of a similar magnitude across MIS3, despite the differing ice extents in the Uk37 reconstruction.

Line 104-105: Which records of Hagemann et al., (2024) suggest that Uk37 is the preferred record? The Uk37 SST record is used as the forcing, so that is circular. Is it the MAR record? If so, I recommend being specific about which records provide support.

Line 115: Is the PATICE reconstruction meant to be shown in Figure 2b?

Version 1:

Reviewer comments:

Reviewer #1

(Remarks to the Author)

I've gone through my initial review, the author's response, and the updated manuscript. Everything looks good to me. I think the authors did a great job addressing everything. Nothing further from me.

(Remarks on code availability)

I looked at the code, but did not "review" it. I'm not a modeller or work with code, so I can't really comment on it. Seemed like the other reviewers didn't find any faults with the code.

Reviewer #2

(Remarks to the Author)

Thank you to the authors for taking on my concerns and comments. I have taken a look through all the edits which have been provided, along with the additions and believe all my concerns have been addressed and I am happy to accept it as it is.

(Remarks on code availability)

Good looks good and is accessible for the build and code needed for the model.

Reviewer #3

(Remarks to the Author)

The manuscript has been revised based on comments from 3 reviewers. My first review primarily focused on how the uncertainty of the two Uk37 temperature reconstructions impacts the glacial index forcing, and subsequently the two PIS reconstructions. I have a few remaining minor comments surrounding this, but otherwise the authors have adequately addressed all of my other points. Thanks to the authors for the detailed responses and discussion. Well done on an excellent manuscript!

Main point:

I agree that the difference between the two PI-LGM anomalies is increased by the presence of millennial-scale variability in the Uk37 record, which then determines whether MIS5 is more LGM-like or more PI-like. This is the key aspect to note, and I'm glad that the authors have included this. However, I disagree that the glacial index is completely independent from the uncertainty in the SST reconstructions. For example: If the 1.5°C difference in PI-LGM anomaly is enough to drive more LGM-like conditions during MIS 5, and the uncertainty within one of the SST reconstructions is on the order of 1.5°C, then it's challenging to distinguish the scenarios based on the face-value of the SST reconstructions. Additionally, the PI interval (average of 0-3ka) is determined by a very small amount of data from Hagemann et al. 2024, which differs by ~1.5°C between the two reconstruction methods. If the PI data is systematically biased by either variability in the calibration, during measurement, or from other environmental factors, this could reproduce very different ice sheet histories.

This all being said, I absolutely agree with the authors that the Uk37 reconstruction is supported based on MAR at the site, other landform markers, and Be10 at a nearby site. I think that adequately addressing this is outside the scope of this paper, and that this difference does not impact the author's main points surrounding insolation and millennial-scale variability. My only recommendation is that lines 152-153:

"The glacial index method and therefore the modelled ice sheet behavior are independent of the uncertainty in the comparison of the temperature records and offsets of both proxies"

Be either removed, or revised to something along the lines of:

"The glacial index method and therefore the modelled ice sheet behavior are influenced by the presence of millennial-scale climate variability present in the MR16-09 PC03-based climate reconstructions."

Such that it notes the influence of the data and supports the importance of millennial-scale variability on the reconstruction.

Minor points:

- 1) Figure 2: I would plot EDML d18O on normal y-direction rather than inverted, such that it qualitatively aligns with SST.
- 2) Line 96: Should “distinction” be “distinct”?
- 3) Line 170: Should “Leeside” be “leeside”?

(Remarks on code availability)

Final Response to Reviewer #1:

Review of “Orbital and millennial-scale forcing of the Patagonian Ice Sheet throughout the Last Glacial Cycle”

Andrés et al present a modeling study of the Patagonian Ice Sheet (PIS) through the last glacial cycle that explores why the PIS reached its maximum likely before the global LGM, and to a lesser extent, why this is out of phase with Northern Hemisphere ice sheets. This study adds to an expansive body of literature over the last several decades that have all focused on 1) when were glaciers/ice sheets in the Southern Hemisphere expanded, 2) when did they retreat, 3) what forced these changes, and 4) how/why is this signature different from the Northern Hemisphere.

I am not an ice-sheet modeler, but I am a Quaternary paleoclimatologist/geochronologist who thinks about these questions above. As such, I found this paper quite interesting and it had me thinking about my own research and sifting through papers and furthermore, I think this manuscript is generally well written. I honestly cannot comment if this manuscript is “correct”, but the methods seem sound, the discussion reasonable, and I do think it a worth addition to the broader discussion surrounding Southern Hemisphere glaciation. I only have a few suggestions and questions:

1) At first, I struggled a bit with “What is the main selling point here?”, which frankly is probably not a good sign for a manuscript like this. I think in the end the main selling points are a) a scheme that uses integrated insolation as a driver of long-term change provides the best data-model fit, and 2) more traditional treatments of insolation are unable to explain the pattern of PIS glaciation. I encourage the authors to really emphasize integrated insolation. For example, its mentioned in the abstract, but the first thing I read (and thought was the main takeaway) was that the PIS advanced during MIS/3...to which my response was “we’ve known that for awhile.”

Response: We are pleased with the positive response and the valuable feedback provided by Reviewer 1. The integrated summer energy enables to explain not only the earlier maximum extension of the PIS to the detriment of more traditional treatment of insolation such as the peak summer insolation, peak winter insolation or summer duration, but also explaining the glacial advance over the MIS4. Additional info to support the integrated summer energy as the best model-data fit has been included in the supplementary information material. Changes to stress the main findings have been implemented in the abstract as suggested.

2) Selection of sediment core MR16-09 PC03. This seems a bit arbitrary, and I do wonder about how reproducible these results are using other temp records. I strongly suspect this record was used because it’s the longest record from the region and spans the last glacial cycle (and probably the most recent record?). But it may be worth exploring how other records affect the results. ODP sites 1233 and 1234 slightly north of the record used here also have alkenone SST records. I believe at least one of those also spans the entirety of the last glacial. How easily can these be incorporated. At the bare minimum, I do not think

these records can be completely ignored....are they different/similar to MR16-09 PC03? Should other records be plotted on a figure as well?

Response: The selection of the sediment core MR16-09 PC03 was due to being the longest and most recent record from the region with a frequency sampling that allows for the study of the PIS variability over the Last Glacial Cycle with thorough accuracy. Additionally, it is located in the center of the former latitudinal extension of the PIS. However, we acknowledge that the discussion of other records is beneficial to enhance the robustness of the findings. The orbital scale variability has imprints not only in MR16-09 PC03 but also in different cores over the South Pacific. The marine records located north of the study zone ODP-1233 and ODP-1234 have imprints of the orbital scale variability, suggesting two main periods of relative minimum temperature: during the MIS4 and late MIS3 and early MIS2, both periods agree in the timings of the orbital-scale variability induced by the ISE. South of MR16-09 PC03, the core MD07-3128 does not cover the MIS4, therefore, limited information can be extracted. Far off-shore records covering the time period here studied reveal that minimum sea-surface temperatures earlier than LGM are captured, however they have a limitation in the sampling frequency. Additionally, EDML also has imprints of such variability. Figures 1 and 2 of the main text were modified, including the location of the offshore records mentioned above and by including the $\delta^{18}O$ record of EDML, respectively. Additionally, a figure with the temperature reconstruction is now shown in the supplemental material (Supplementary Figure 2). An earlier study (Castillo-Llarena et al., 2024) has shown have concluded that the timings of maximum glacial advance and the consequent asynchronicity with the global mean behaviour are not captured by Antarctic cores, and local proxy data are needed to capture the local variability (Castillo-Llarena et al., 2024). However, imprints of the orbital-scale variability are indeed captured. A discussion on the imprints in other offshore records in the southeastern Pacific Ocean has been included.

3) Like #2, I find it hard to believe I just read an entire Southern Hemisphere glaciation paper without seeing a single ice-core record plotted, let alone used. The authors routinely mention that “millennial-scale Antarctic cold stadials played a critical role....” I think the authors need to at the bare minimum show these on a figure somewhere. If I’m playing devil’s advocate, what would happen if you simulated the last glacial cycle using a temp record from Antarctica, or at least WAIS (realizing that WAIS does not go back 140ka)? Has this been done before? Maybe what I’m also getting at here is how do the authors simulations compare to previous efforts. Since this is Nature Communications I think you have a little more leeway with article length.

Response: As commented in the response of #2, insights of the Orbital Scale variability are being visible not only in the core initially considered in this study but also in further offshore records analysed and the Antarctic ice core record EDML. We thank the reviewer for this suggestions that help the interpretation of the results. The $\delta^{18}O$ record from EDML and the Antarctic Isotope Maxima are now shown in Figure 2. The influence of the millennial-scale variability on the core record MR1609 PC03 has been largely discussed by Hagemann et al. (2024), and it has been highlighted now in Figure 2 to facilitate the discussion. Previous modelling efforts focusing on the PIS have shown that the Antarctic cores failed to reproduce the timing of the maximum extent of Patagonia during the late MIS3, highlighting the

importance of considering local records to capture the variability of the PIS with enough accuracy (Castillo-Llarena et al., 2024).

4) I like how the authors made a decent attempt to bring in the geologic record. I think many modeling groups often don't even bother, so this is a nice touch. My only comment on this aspect would be how the authors evaluate model-data success or mismatches. There are a lot of references to how the author's efforts *broadly* do a good job at reproducing the geological record of PIS change, but it is a bit tough to gauge from the text and figures. I will say that having the 35-ka outline in Figure 3 is very helpful although I was wondering a bit about the slight mismatch between model output and the geologic record at 35 ka. Might be nice for the authors to comment on that. Also, why does the model output appear shifted east at 35 ka? Or maybe another way to put it is that it undershoots ice on the west side but overshoots on the east side. I think this paper would be stronger if the comparison between model output and the geological record were a bit more robust, maybe there is some way to quantify this. The most obvious targets/benchmarks are MIS 4 (PIS at max near max as constrained by geologic record) and MIS 2 where there are numerous dated moraines.

Response: We are pleased with the positive feedback provided by Reviewer 1 with regard to the geologic records. On the one hand, our simulations are able to reproduce the timing of the two main periods of glacial advance proposed by the available literature, during the MIS4 and during the late MIS3. On the other hand, our simulations are also capable of simulating a maximum ice extent earlier than the LGM. Despite MPI-ESM standing as the best-performing climate model (Yan et al., 2022; Castillo-Llarena et al., 2024), mismatches between the reconstructed and modeled ice extent have been found not only in this research work but also in previous works, being largely attributed to the coarse resolution of the atmospheric forcing and the consequent prescribed topography (Castillo-Llarena et al., 2024). Coarse and inaccurate orographic forcing in Patagonia induces wetter and colder temperatures over the leeside of the Andes (Damseaux et al., 2019) promoting the ice extent beyond its geologically reconstructed margin. We have compared the total modelled area with the existing PIS geochronological reconstruction PATICE (Davies et al., 2020) showing a good agreement despite the mismatch with the geological shape due to the atmospheric forcing of the paleoclimate model used. The total area of PATICE for each time slide has been added to figure 2 for comparison.

References:

- Castillo-Llarena, A., Retamal-Ramírez, F., Bernales, J., Jacques-Coper, M., Prange, M., & Rogozhina, I. (2024). Climate and ice sheet dynamics in Patagonia throughout marine isotope stages 2 and 3. *Climate of the Past*, 20(7), 1559-1577.
- Damseaux, A., Fettweis, X., Lambert, M., & Cornet, Y. (2019). Representation of the rain shadow effect in Patagonia using an orographic-derived regional climate model. *International Journal of Climatology*.
- Davies, B. J., Darvill, C. M., Lovell, H., Bendle, J. M., Dowdeswell, J. A., Fabel, D., ... & Thorndycraft, V. R. (2020). The evolution of the Patagonian Ice Sheet from 35 ka to the present day (PATICE). *Earth-Science Reviews*, 204, 103152.

Hagemann, J. R., Lamy, F., Arz, H. W., Lembke-Jene, L., Auderset, A., Harada, N., ... & Tiedemann, R. (2024). A marine record of Patagonian ice sheet changes over the past 140,000 years. *Proceedings of the National Academy of Sciences*, 121(12), e2302983121.

Yan, Q., Wei, T., & Zhang, Z. (2022). Modeling the climate sensitivity of Patagonian glaciers and their responses to climatic change during the global last glacial maximum. *Quaternary Science Reviews*, 288, 107582.

Final Response to Reviewer #2:

This manuscript is one of the first to attempt to model the Patagonian Ice Field / Ice Sheet through the Last Glacial Period using a new ocean core that provides an alkenone-based sea surface temperature record (MR16-09 PC03) from Hagemann et al. (2024). The ability to use such a record is really interesting and fascinating, as such attempts have previously not been able to do so. I do commend the manuscript for attempting to do so and being at the forefront of modelling the Patagonian Ice Field using such recent and new data.

Noteworthy results

- Modelling the Patagonian Ice Field through the Last Glacial Period itself is noteworthy
- Multiple advances of the Ice Field both within MIS 4 and 3 that are varied due to inter-millennial variability
- That, using the records provided, the simulations agree with PATICE (Davies et al., 2020), but disagree with a smooth deglaciation but rather varied deglaciated with interannual fluctuations
- Summer and winter isolation do not provide a clear explanation for ice variability, but more closely follow summer energy.

Does the work support the conclusions?

The evidence provided in Figure 2, the model output, and the comparison with other records provides clear evidence for the conclusions presented within the paper. Would have been nice to have seen more of a comparison with more outlines of the Patagonian Ice Sheet, to see it through different timesteps, and if the model with the new SST record could match more than just the 35ka transition.

Response: We are pleased with the positive feedback provided by Reviewer 2. As suggested, PATICE total area has been incorporated into Figure 2 for comparison. Additional figures showing our model results and its comparison with PATICE at 35, 30, 25, 20 and 15 ka outlines with both proxies (supplementary figures 4 and 5). Moreover, figures showing the ice sheet velocities for 80, 65, 57, 35 and 21 ka can now be found in supplementary figure 3.

Flaws in data analysis:

While the output itself of the ice extent and thickness are valuable in and of themselves, I think more could be said from your model which you have negated to do so. It would be really interesting and useful to understand what is happening with the velocity fields, how the ice is evacuating itself through the ice sheet. Does this change as the ice sheet involves through the Last Glacial Period? Is the thickness realistic, and does it match what other

models have done over Patagonia for the LGM for example? While it may capture 35ka well, does it actually match other time periods well?

Response: We thank the reviewer for this thoughtful and constructive comment. Numerical reconstructions of the Ice Thickness of the Patagonian Ice Sheet diverge considerably, being a matter of debate between the existing literature. Ice sheet thickness estimations are ranging between 1500 and 3200 m (Sudgen et al., 2002; Yan et al., 2022; Wolf et al., 2023; Castillo-Llarena et al., 2024). Those discrepancies could be attributed to model resolution, input data and parametrizations. Our results regarding the ice sheet thickness are within the existing range of values proposed by previous studies. Additional figures showing the comparison between our model simulations and the PATICE reconstruction at 35, 30, 25, 20 and 15 ka have been included (Supplementary Figure 4). Figures with the ice velocity for the same time slices are included in the Supplemental Material (Supplementary Figure 5). The modeled PIS is mainly drained by fast-flowing outlet glaciers to the east, while even larger velocities can be found in the western margin (Supplementary Figure 3 and 5). The velocities of ice streams that terminate in the Pacific Ocean range over 1000 m/year, whereas the velocities of land-terminating fast-flowing outlet glaciers at the eastern margins of the PIS are generally within 100 m/year. The ice divide mainly extends north-south along the Patagonian Andes between. Variations in the ice velocities can be found throughout the different time steps, particularly focused in the margins of the ice sheet (Supplementary Figure 3 and 5).

Methodology:

This modelling is a continuation of a modelling studies from the same authors over Patagonia (Castillo-Llarena et al., 2024). While much of the methodology is similar, there are certainly some differences which should be noted in the methodology. For example, how the input data was made and set up for the model. What are the parameters used in the model? Any differences from the default parameter values? This could be in the form of a table so we know the parameters varied, as from reading the former paper before this, there is also no clear indication of parameter values. Also, if there was any additional parameters or modules used to enable the model to be ran through to completion. At the moment, I do not think there is enough for replication purposes.

Response: The model setup is the same as in Castillo-Llarena et al. (2024). The parameters used to perform the simulations shown in this paper are now included in the supplemental material of the manuscript (Supplementary table 1). More detailed description of the model configuration has been included into the method section of the revised manuscript.

Attached is also a commented version of the manuscript for minor points.

Response: All comments have been addressed.

Overall comment and review verdict:

I personally think the amount of information that could be extracted from such a study is not inline with the reduced amount of information a nature paper allows the author. While the

study and findings are some of the first to have been conducted over Patagonia, the outputs need to be described in-depth, with further information also to be presented in the methodology. The study itself is highly commendable and a pleasure to read, but I do not see Nature Comms being the best fit for such a paper. If the authors wish to resubmit to another journal, I would be more than happy to be put down as a reviewer to provide further feedback.

We sincerely thank the reviewer for their thoughtful and encouraging feedback. We are particularly grateful for the kind remarks regarding the novelty of our work over Patagonia, as well as the acknowledgment that the study was a pleasure to read and scientifically commendable. We remain confident that Nature Communications is a suitable venue for this work due to its relevance to global climate processes, the originality of the dataset, and the implications for understanding ice sheet behavior outside polar regions. We believe that the main message of the paper can be effectively conveyed within the constraints of Nature Communications format.

References

- Castillo-Llarena, A., Retamal-Ramírez, F., Bernales, J., Jacques-Coper, M., Prange, M., & Rogozhina, I. (2024). Climate and ice sheet dynamics in Patagonia throughout marine isotope stages 2 and 3. *Climate of the Past*, 20(7), 1559-1577.
- Wolff, I. W., Glasser, N. F., Harrison, S., Wood, J. L., & Hubbard, A. (2023). A steady-state model reconstruction of the patagonian ice sheet during the last glacial maximum. *Quaternary Science Advances*, 12, 100103.
- Sugden, D. E., Hulton, N. R., & Purves, R. S. (2002). Modelling the inception of the Patagonian icesheet. *Quaternary International*, 95, 55-64.
- Yan, Q., Wei, T., & Zhang, Z. (2022). Modeling the climate sensitivity of Patagonian glaciers and their responses to climatic change during the global last glacial maximum. *Quaternary Science Reviews*, 288, 107582.

Final Response to Reviewer #3:

This study produces a new reconstruction of the Patagonian Ice Sheet (PIS) across the last glacial cycle, extending a previous reconstruction which largely covered MIS3. The authors relate changes in their modeled PIS to integrated summer insolation and millennial-scale climate variability. The ice sheet reconstruction is forced by a continuous SST record over the past 140kya from the eastern Pacific offshore of Chile, which records variability in the western extent of the PIS. This provides insight into regional climate dynamics over a significantly longer period than previous nearby records. This SST record is used to construct a glacial index which guides the interpolation of glacial and pre-industrial climate states to be used as model forcing.

The results of this study provide important constraints on the history of the PIS beyond what was previously available and reveals new millennial-scale dynamics within previously

studied intervals. I congratulate the authors on the work that went into this study, and it will undoubtedly provide a great resource to gain further insight into the PIS.

My main points of discussion focus on the capability to meaningfully differentiate between the two forcings across MIS5, which in my view is weakened by the lack of independent verification and uncertainty in the forcing. I will discuss this below, along with the authors three primary findings which I take directly from the abstract:

(1) “Our analysis suggests PIS advances during the Marine Isotope Stage (MIS) 4 and late MIS3...” This finding is dependent on the absolute temperature of the Uk37 record, which is the primary difference, rather than high-frequency variability, between the two forcings across MIS5.

(2) “We show that millennial-scale climate variability played a role in the PIS dynamics...” This is an important observation for this study, as it shows that the model can sustain a dynamic PIS in-phase with millennial-scale climate variability.

(3) “...while the long-term PIS evolution can be attributed to changes in the integrated summer insolation” This is also an important observation. However, it is unclear if this observation is limited to this reconstruction or is a consistent feature across all of the Southern Hemisphere.

Overall, this study provides valuable new information surrounding the history of the PIS. A strength of this study is the identification of millennial-scale PIS variability across MIS3, which warrants further description. In my view, this paper should provide more conservative claims surrounding the MIS5-4 history and additional insight into the dynamics of newly observed millennial-scale variations.

Response: We sincerely thank the reviewer for their thoughtful and constructive comments, as well as for their positive assessment of our study. We are particularly grateful for the recognition of the effort invested in developing this new reconstruction of the Patagonian Ice Sheet and for highlighting the value of our work in extending the temporal framework and identifying orbital and millennial-scale variability. We appreciate the reviewer’s suggestions regarding a more cautious interpretation of the MIS5-4 interval and the need for further elaboration on the dynamics observed during MIS3. In response, we have revised the relevant sections to better reflect the uncertainties associated with MIS5-4 and have expanded our discussion of the millennial-scale signals during MIS3, as detailed below.

Main Discussion Points:

1) My main discussion point is the impact of the of glacial index in the PIS reconstruction. This is particularly evident in the difference in extent during MIS5, where the reconstruction based on Uk’37 rapidly expands at ~115 kya. I presume this is due to the systematically lower temperatures in Uk’37 as compared to Uk37.

How well understood is the temperature uncertainty using these SST reconstruction methods? Neither this study, nor the data source Hagemann et al., 2024, discuss this. The

difference between records is $\sim 1.25\text{--}1.50^\circ\text{C}$, which is in the range of absolute uncertainty in comparable reconstruction methods. Based on the uncertainty in the SST's, can the different PIS reconstruction scenarios be differentiated?

While I agree with the author's interpretation that other lines of evidence (primarily the MAR/biomarker data of Hagemann et al., 2024 and the glacial advancement markers cited within Hagemann et al., 2024) suggest that the PIS reconstruction based on Uk'37 can be excluded, I feel that discussion of uncertainty within the SST forcing and the sensitivity of the model to that forcing is a necessary discussion in order to justify an advancement across MIS4 rather than MIS5. As it stands, the results only suggest that PIS initiation occurred by $\sim 115\text{ka}$ and reached its approximate LGM extent by at least the onset of MIS4.

Response: The simulations have been performed by using a transient forcing derived from a glacial index method. The method takes as reference values the pre-industrial conditions and the Last Glacial Maximum (LGM, 23,000 to 19,000 years before present). During the LGM, the average temperature of Uk'37 is around 1.5°C warmer than for Uk37. Such a difference is a consequence of the millennial-scale variability captured by Uk37, inducing much more extreme conditions over the LGM and the late MIS3. During MIS5, the temperatures derived from Uk'37 are close to those derived by the core during the LGM, therefore, the forcing during those periods mimics the LGM forcing. The glacial index method and therefore the modelled ice sheet behaviour are independent of the uncertainty in the comparison of the temperature records and the offsets of both proxies.

2) For the observed millennial-scale variations in extent and volume, where do these changes manifest? I feel this is key source of information which is not discussed. Are these changes in extent isolated to specific regions, or are they well distributed across the PIS? This could be included as a supplemental spatial anomaly map.

How do the changes in this new reconstruction compare to other reconstructions like PATICE or that of Darvill et al., 2016? Does the difference in extent in Figure 3 in the 35ka panels imply that this reconstruction underestimates the geological evidence?

Response: Our simulations are performed using the glacial index method, which enables generating climate states at any given time through the weights interpolation of the LGM and PI snapshots of the best-fitted paleoclimate model in Patagonia based on previous modelling studies (Yan et al., 2022; Castillo-Llarena et al., 2024). The imprints of both the orbital and the millennial-scale variability are present in the core record used to create the glacial index curve. While we reckon the limitations of the method which assumes that the perturbations are applied homogeneously over the full study zone, the method has been largely used for paleoclimate studies in Patagonia and elsewhere (eg., Charbit et al., 2007; Niu et al., 2019, Niu et al., 2021; Mas e Braga et al., 2021; Castillo-Llarena et al., 2024). The marks of the millennial-scale variability are present in the vicinity of the offshore records, in the western and eastern margins of the PIS, particularly north of 44°S and south of 50°S (Supplementary Figure 6).

The obtained geometry of the maximum extension of PIS is then closely linked with the near surface air temperature of the cold member of the glacial index, in this case, the full LGM

conditions of the MPI-ESM1-2-LR, which due to the coarse atmospheric model resolution and the topographic misrepresentation leads to mismatches with the geochronologically reconstructed geometry of the PIS (Castillo-Llarena et al., 2024). Matching the simulated ice sheet with the geochronological reconstruction under different timesteps is not the main focus of this study but to understand the overall evolution of the ice mass over the last Glacial Cycle and why its asynchronous behaviour with the northern hemisphere ice sheets.

3) In a recent study (Castillo-Llarena et al., 2024), Antarctic ice core data is used as a forcing over the last 70 kya. Has this been done for comparison over the last glacial cycle here? This could provide a necessary source of independent verification for the orbital-scale variability of the PIS.

The structure and magnitude across MIS3 is also consistent between the EDC forced reconstruction of Castillo-Llarena et al., 2024 and Uk37 shown here. Including a comparison between the two could provide insight into their similarities and differences across both millennial and orbital timescales. Furthermore, how do the other reconstructions of Castillo-Llarena et al., 2024 which utilize other SST records across MIS3/2 compare to the ones presented here? If the PIS advancement during late MIS3 is a regional phenomenon, is it expressed across the wider SST forcings and in their respective ice sheet reconstructions?

Response: The core MR16-09 PC03 was selected due to being the longest and most recent record from the regions with a frequency sampling that allows the study of the variability over the Last Glacial Cycle accurately. However, we acknowledge that the discussion of other records and moreover those used in previous modelling efforts is beneficial in enhancing the discussion and robustness of the findings. Existing modelling efforts of the PIS have concluded that the timings of maximum glacial advance and the consequent asynchronicity with the global mean behaviour is not captured by Antarctic cores, and local proxy data is needed to capture the local variability (Castillo-Llarena et al., 2024). However, imprints of the orbital-scale variability are indeed captured. An analysis comparing different SST reconstructions based on other off-shore records over the Last Glacial Cycle was included. Also, the results of Castillo-Llarena et al. (2024) have been discussed. The orbital-scale variability has imprints not only in MR16-09 PC03 but also in different cores over the south Pacific (Supplementary Figure 2). The near offshore records located north of the study zone ODP-1233 and ODP-1234 have imprints of the orbital scale variability, suggesting two main periods of relative minimum temperature: during the MIS4 and during late MIS3 and early MIS2, both periods agree in the timings of the orbital-scale variability induced by the ISE. South of MR16-09 PC03, the core MD07-3128 does not cover the MIS4, therefore, limited information can be extracted. The far offshore records GeoB3327-5 and PS75-034/2, despite their limitation in the sampling frequency, reveal that minimum sea-surface temperatures earlier than LGM are captured with a variability in line with the ISE. Additionally, EDC also has imprints of such variability.

4) In a similar vein, is statement in lines 136-138: “The local minimum in integrated summer energy during the beginning of the MIS4 aligns well with the timing of the PIS growth, while the positive anomaly in early MIS3 helps explain the gradual disintegration of the ice sheet (Fig. 2b).” further supported by previous reconstructions of the PIS?

Response: Periods of glacial advance are broadly in line with glacial advance suggested by previous research efforts (Hagemann et al., 2024; Sproson et al., 2024). Additional independent verification of the timings can be found based on geological records for Patagonia (Moreno et al., 2015; Garcia et al., 2018; Garcia et al., 2021; Glasser et al., 2011; Hein et al., 2010; Kaplan et al., 2008; Peltier et al., 2021; Mendelova et al., 2020) and the southern hemisphere mid latitudes outside Patagonia (Schaefer et al., 2015; Hall et al., 2020; Rudolph et al., 2020; Lesic et al., 2022). Additional offshore records have been included in the supplemental material.

5) Does this ice sheet reconstruction support glacial sediment delivery to the site? The western margin is reduced as compared to previous reconstructions, with more significant advances across the eastern margin. Does this reduced western extent still result in marine terminating conditions, or ice extent across the continental margin as suggested by Hagemann et al? In other words, is the PIS reconstruction presented here consistent with the conditions necessary to reproduce the sediment forcing record?

Response: To isolate the atmospheric dynamics and the ocean dynamics, the model is set in a way that as soon as it reaches the coast, it calves. Our model outputs are therefore considering a simple calving scheme. However, we interpret the timings of advance and retreat of the ice in agreement with Hagemann et al. (2024), but also with the geologically constrained margin of the PIS. Moreover, our model simulations are forced by climate model input that due to its coarse resolution and the poorly constrained topography are unable to capture the climate variability with enough proficiency to enable us to reproduce in a fully realistically way the geochronologically reconstructed margins and the sediments forcing record.

Minor Comments:

Line 87-88: References Figure 2e here.

Done

Line 99-101: Does the Be10/Be9 contradict the results of PIS advancement at the start of MIS4? The Be10/Be9 is indicative of terrigenous sedimentation beginning at ~80kya, with oscillations preceding it. The record across MIS5 is of a similar magnitude across MIS3, despite the differing ice extents in the Uk37 reconstruction.

Response: Sproson et al. (2024) have used Be10/Be9 as an indicator of the glacial variation of the PIS, identifying two main periods of glacial advance. The periods broadly align with our timings of the glacial advance of both integrated summer insolation and MR19-09 PC03. The age model uncertainties increase particularly during the proposed glacial advance around the MIS4. This has been included in the new manuscript.

Line 104-105: Which records of Hagemann et al., (2024) suggest that Uk37 is the preferred record? The Uk37 SST record is used as the forcing, so that is circular. Is it the MAR record? If so, I recommend being specific about which records provide support.

We referred to the MAR record. This was corrected in the manuscript.

Line 115: Is the PATICE reconstruction meant to be shown in Figure 2b?
PATICE reconstruction has been shown for comparison

References

- Castillo-Llarena, A., Retamal-Ramírez, F., Bernales, J., Jacques-Coper, M., Prange, M., & Rogozhina, I. (2024). Climate and ice sheet dynamics in Patagonia throughout marine isotope stages 2 and 3. *Climate of the Past*, 20(7), 1559-1577.
- Charbit, S., Ritz, C., Philippon, G., Peyaud, V., & Kageyama, M. (2007). Numerical reconstructions of the Northern Hemisphere ice sheets through the last glacial-interglacial cycle. *Climate of the Past*, 3(1), 15-37.
- García, J. L., Lüthgens, C., Vega, R. M., Rodés, Á., Hein, A. S., & Binnie, S. A. (2021). A composite 10 Be, IR-50 and 14 C chronology of the pre-Last Glacial Maximum (LGM) full ice extent of the western Patagonian Ice Sheet on the Isla de Chiloé, south Chile (42° S). *E&G Quaternary Science Journal*, 70(1), 105-128.
- Glasser, N. F., Jansson, K. N., Goodfellow, B. W., De Angelis, H., Rodnight, H., & Rood, D. H. (2011). Cosmogenic nuclide exposure ages for moraines in the Lago San Martín Valley, Argentina. *Quaternary Research*, 75(3), 636-646.
- Hagemann, J. R., Lamy, F., Arz, H. W., Lembke-Jene, L., Auderset, A., Harada, N., ... & Tiedemann, R. (2024). A marine record of Patagonian ice sheet changes over the past 140,000 years. *Proceedings of the National Academy of Sciences*, 121(12), e2302983121.
- Hall, B. L., Lowell, T. V., & Brickle, P. (2020). Multiple glacial maxima of similar extent at 20–45 ka on Mt. Osborne, East Falkland, South Atlantic region. *Quaternary Science Reviews*, 250, 106677.
- Hein, A. S., Hulton, N. R., Dunai, T. J., Sugden, D. E., Kaplan, M. R., & Xu, S. (2010). The chronology of the Last Glacial Maximum and deglacial events in central Argentine Patagonia. *Quaternary Science Reviews*, 29(9-10), 1212-1227.
- Kaplan, M. R., Fogwill, C. J., Sugden, D. E., Hulton, N. R. J., Kubik, P. W., & Freeman, S. P. H. T. (2008). Southern Patagonian glacial chronology for the Last Glacial period and implications for Southern Ocean climate. *Quaternary Science Reviews*, 27(3-4), 284-294.
- Lešić, N. M., Streuff, K. T., Bohrmann, G., & Kuhn, G. (2022). Glacimarine sediments from outer Drygalski Trough, sub-Antarctic South Georgia—evidence for extensive glaciation during the Last Glacial Maximum. *Quaternary Science Reviews*, 292, 107657.
- Mas e Braga, M., Bernales, J., Prange, M., Stroeven, A. P., & Rogozhina, I. (2021). Sensitivity of the Antarctic ice sheets to the warming of marine isotope substage 11c. *The Cryosphere*, 15(1), 459-478.
- Mendelová, M., Hein, A. S., Rodés, Á., & Xu, S. (2020). Extensive mountain glaciation in central Patagonia during Marine Isotope Stage 5. *Quaternary Science Reviews*, 227, 105996.
- P. I. Moreno et al., Radiocarbon chronology of the last glacial maximum and its termination in northwestern Patagonia. *Quaternary Science Reviews* 122, 233-249 (2015).
- Niu, L. U., Lohmann, G., Hinck, S., Gowan, E. J., & Krebs-Kanzow, U. T. A. (2019). The sensitivity of Northern Hemisphere ice sheets to atmospheric forcing during the last glacial cycle using PMIP3 models. *Journal of Glaciology*, 65(252), 645-661.

Niu, L., Lohmann, G., Gierz, P., Gowan, E. J., & Knorr, G. (2021). Coupled climate-ice sheet modelling of MIS-13 reveals a sensitive Cordilleran Ice Sheet. *Global and Planetary Change*, 200, 103474.

Peltier, C., Kaplan, M. R., Birkel, S. D., Soteres, R. L., Sagredo, E. A., Aravena, J. C., ... & Schaefer, J. M. (2021). The large MIS 4 and long MIS 2 glacier maxima on the southern tip of South America. *Quaternary Science Reviews*, 262, 106858.

Rudolph, E. M., Hedding, D. W., Fabel, D., Hodgson, D. A., Gheorghiu, D. M., Shanks, R., & Nel, W. (2020). Early glacial maximum and deglaciation at sub-Antarctic Marion Island from cosmogenic ^{36}Cl exposure dating. *Quaternary Science Reviews*, 231, 106208.

Sproson, A. D., Yokoyama, Y., Miyairi, Y., Aze, T., Clementi, V. J., Riechelsohn, H., ... & Childress, L. B. (2024). Near-synchronous Northern Hemisphere and Patagonian Ice Sheet variation over the last glacial cycle. *Nature Geoscience*, 17(5), 450-457.

Sugden, D. E., Hulton, N. R., & Purves, R. S. (2002). Modelling the inception of the Patagonian icesheet. *Quaternary International*, 95, 55-64.

Wolff, I. W., Glasser, N. F., Harrison, S., Wood, J. L., & Hubbard, A. (2023). A steady-state model reconstruction of the patagonian ice sheet during the last glacial maximum. *Quaternary Science Advances*, 12, 100103.

Yan, Q., Wei, T., & Zhang, Z. (2022). Modeling the climate sensitivity of Patagonian glaciers and their responses to climatic change during the global last glacial maximum. *Quaternary Science Reviews*, 288, 107582.

Final Response to Reviewer #1

I've gone through my initial review, the author's response, and the updated manuscript. Everything looks good to me. I think the authors did a great job addressing everything. Nothing further from me.

Reviewer #1 (Remarks on code availability):

I looked at the code, but did not "review" it. I'm not a modeller or work with code, so I can't really comment on it. Seemed like the other reviewers didn't find any faults with the code.

Response: We sincerely thank Reviewer #1 for the positive feedback.

Final Response to Reviewer #2

Thank you to the authors for taking on my concerns and comments. I have taken a look through all the edits which have been provided, along with the additions and believe all my concerns have been addressed and I am happy to accept it as it is.

Reviewer #2 (Remarks on code availability):

Code looks good and is accessible for the build and code needed for the model.

Response: We sincerely thank Reviewer #2 for the positive feedback. We are pleased that the revisions and additions have satisfactorily addressed all concerns.

Final Response to Reviewer #3

The manuscript has been revised based on comments from 3 reviewers. My first review primarily focused on how the uncertainty of the two Uk37 temperature reconstructions impacts the glacial index forcing, and subsequently the two PIS reconstructions. I have a few remaining minor comments surrounding this, but otherwise the authors have adequately addressed all of my other points. Thanks to the authors for the detailed responses and discussion. Well done on an excellent manuscript!

Response: We sincerely thank Reviewer #3 for the feedback. We have implemented all changes suggested by the reviewer.

Main point:

I agree that the difference between the two PI-LGM anomalies is increased by the presence of millennial-scale variability in the Uk37 record, which then determines whether MIS5 is more LGM-like or more PI-like. This is the key aspect to note, and I'm glad that the authors have included this. However, I disagree that the glacial index is completely independent from the uncertainty in the SST reconstructions. For example: If the 1.5°C difference in PI-LGM anomaly is enough to drive more LGM-like conditions during MIS 5, and the uncertainty

within one of the SST reconstructions is on the order of 1.5°C, then it's challenging to distinguish the scenarios based at the face-value of the SST reconstructions. Additionally, the PI interval (average of 0-3ka) is determined by a very small amount of data from Hagemann et al. 2024, which differs by ~1.5°C between the two reconstruction methods. If the PI data is systematically biased by either variability in the calibration, during measurement, or from other environmental factors, this could reproduce very different ice sheet histories.

This all being said, I absolutely agree with the authors that the Uk37 reconstruction is supported based on MAR at the site, other landform markers, and Be10 at a nearby site. I think that adequately addressing this is outside the scope of this paper, and that this difference does not impact the author's main points surrounding insolation and millennial-scale variability.

Response: We thank the reviewer for this thoughtful and constructive remark. We acknowledge that the limited PI data, and potential systematic biases in reconstructions, add to this uncertainty. As the reviewer notes, however, a full quantification of these uncertainties and their implications for ice-sheet histories lies beyond the scope of the present study. We are reassured that the reviewer concurs that our main conclusions, particularly regarding the role of insolation and millennial-scale variability, remain robust and well supported by the independent lines of evidence considered.

My only recommendation is that lines 152-153:

"The glacial index method and therefore the modelled ice sheet behavior are independent of the uncertainty in the comparison of the temperature records and offsets of both proxies"

Be either removed, or revised to something along the lines of:

"The glacial index method and therefore the modelled ice sheet behavior are influenced by the presence of millennial-scale climate variability present in the MR16-09 PC03-based climate reconstructions."

Such that it notes the influence of the data and supports the importance of millennial-scale variability on the reconstruction.

Response: Thanks for those thoughtful comments. The sentence, as suggested, was removed.

Minor points:

1) Figure 2: I would plot EDML d18O on normal y-direction rather than inverted, such that it qualitatively aligns with SST.

Response: Corrected. Figure has been modified accordingly.

2) Line 96: Should "distinction" be "distinct"?

Response: Corrected.

3) Line 170: Should "Leeside" be "leeside"?

Response: Corrected.